# Masculinization of Red Tilapia (*Oreochromis* spp.) Using 17α-Methyltestosterone-Loaded Alkyl Polyglucosides Integrated into Nanostructured Lipid Carriers

**DOI:** 10.3390/ani13081364

**Published:** 2023-04-16

**Authors:** Jakarwan Yostawonkul, Sirikorn Kitiyodom, Kittipat Supchukun, Nutthanit Thumrongsiri, Nattika Saengkrit, Komkiew Pinpimai, Amin Hajitou, Kim D. Thompson, Kasem Rattanapinyopituk, Masashi Maita, Manoj Tukaram Kamble, Teerapong Yata, Nopadon Pirarat

**Affiliations:** 1The International Graduate Course of Veterinary Science and Technology (VST), Faculty of Veterinary Science, Chulalongkorn University, Bangkok 10330, Thailand; 2National Nanotechnology Center (NANOTEC), National Science and Technology Development Agency (NSTDA), Pathumthani 12120, Thailand; 3Wildlife, Exotic and Aquatic Animal Pathology Research Unit, Department of Pathology, Faculty of Veterinary Science, Chulalongkorn University, Bangkok 10330, Thailand; 4Aquatic Resources Research Institute, Chulalongkorn University, Bangkok 10330, Thailand; 5Cancer Phagotherapy, Department of Brain Sciences, Imperial College London, London W12 0NN, UK; 6Moredun Research Institute, Penicuik EH26 0PZ, UK; 7Laboratory of Fish Health Management, Department of Marine Biosciences, Tokyo University of Marine Science and Technology, Konan 4-5-7, Minato, Tokyo 108-8477, Japan; 8Unit of Biochemistry, Department of Physiology, Faculty of Veterinary Science, Chulalongkorn University, Bangkok 10330, Thailand

**Keywords:** lipid-based nanoparticles, 17 alpha-methyltestosterone, masculinization, red tilapia (*Oreochromis* spp.)

## Abstract

**Simple Summary:**

Lipid-based nanocarriers have proved effective for drug and hormone delivery. The androgen hormone 17 alpha-methyltestosterone (MT) is used for sex reversal in tilapia; however, the hormone has biopharmaceutical limitations such as insolubility, low stability, limited intestinal absorption, and poor bioavailability. Incorporating MT into lipid-based nanocarriers to improve hormone uptake in tilapia is an interesting prospect. In the present study, we reported an improved version of alkyl polyglucosides loaded with 17 alpha-methyltestosterone integrated into nanostructured lipid carriers (APG-NLC) for the masculinization of red tilapia (*Oreochromis* spp.) via oral administration as a feed additive. We demonstrated that nanotechnology-based drug delivery systems represent a novel and promising strategy for improving the oral absorption of these androgen hormones, resulting in a 50% *w*/*w* reduction in the hormonal dosage during the sex reversal period. Therefore, we strongly believe that this newly optimized nanotechnology is one of the solutions for sustainable aquaculture production.

**Abstract:**

The aim of the present study was to optimize a masculinization platform for the production of all-male red tilapia fry by oral administration of 30 and 60 ppm of MT and alkyl polyglucoside nanostructured lipid carriers (APG-NLC) loaded with MT, respectively, for 14 and 21 days. The characterization, encapsulation efficiency and release kinetics of MT in lipid-based nanoparticles were assessed in vitro. The results showed that the MT-loaded nanoparticles were spherical, ranging from 80 to 125 nm in size, and had a negative charge with a narrow particle distribution. The APG-NLC loaded with MT provided higher physical stability and encapsulation efficacy than the NLC. The release rate constants of MT from MT-NLC and MT-APG-NLC were higher than those of free MT, which is insoluble in aqueous media. There was no significant difference in survival between the fish administered MT or the those fed orally with MT-APG-NLC fish. According to the logistic regression analysis, the sex reversal efficacy of MT-APG-NLC (30 ppm) and MT (60 ppm), resulted in significantly higher numbers of males after 21 days of treatment compared with the controls. The production cost of MT-APG-NLC (30 ppm) after 21 days of treatment was reduced by 32.9% compared with the conventional MT treatment group (60 ppm). In all the treatments, the length–weight relationship (LWR) showed negatively allomeric growth behavior (*b* < 3), with a relative condition factor (K_n_) of more than 1. Therefore, MT-APG-NLC (30 ppm) would seem to be a promising, cost-effective way to reduce the dose of MT used for the masculinization of farmed red tilapia.

## 1. Introduction

Red tilapia (*Oreochromis* spp.) is recognized as an important farmed fish species worldwide because it can easily adapt to the environment and has a fast growth rate [1,2]. However, growth retardation at the onset of sexual maturity and overpopulation by excessive or unwanted reproduction are the major concerns for mixed-sex red tilapia production. Male red tilapia are generally preferred by fish farmers over females because of their faster growth and lower feed conversion ratio, resulting in higher profit and better color [3]. To overcome this problem, sex reversal using 17 alpha-methyltestosterone (MT) is commonly used in red tilapia culture globally to produce all-male fry [4,5]. It is added to the feed at 30–60 mg/kg of the diet, which is then fed to fish daily for 21–60 days post-hatching (dph), resulting in over 90% male sex reversal [6,7]. The effective concentration of MT ranges from 10 µg/L to 1000 µg/L and is administered as a sex reversal treatment during embryogenesis. Furthermore, a sex reversal effect from female to male has been observed in more than 80% of red tilapia treated with the hormone [8]. Although MT has been proven to be an effective hormone, it is a hydrophobic compound with limited intestinal absorption, low bioavailability and the potential to precipitate in the rearing water and the environment [9,10]. The consequential effects of MT on the environment and the consumer are increasing concerns for the public.

Recently, lipid-based nanoparticles (LBN) have been developed and used for drug and hormone delivery [11,12]. The first generation of LBN tended to consist of two types of nanoparticles: solid lipid nanoparticles (SLN) and nanoemulsions (NE) [13]. SLN and NE are typically synthesized with solid and liquid lipids as a lipid core surrounded by a surfactant. The nanostructured lipid carriers (NLC) are an improved version of lipid-based nanocarriers that consist of a mixture of solid and liquid lipids in different ratios, with particle sizes ranging from 20 to 200 nm [14]. The physicochemical characteristics of LBN, such as particle size, shape and charge, can be controlled through the external energy and the components of the formulation. Furthermore, the particles’ surface can be modified to induce the specificity of the target site and to regulate the kinetic release of the hormone [15]. LBN have gained much interest for their low toxicity, high bioavailability and drug solubility when incorporated in nanoparticles to allow the delivery of hydrophobic drugs or lipid compounds to the target sites [16,17,18,19,20]. They offer several benefits, including low cost, large-scale production, high loading capacity and high stability and versatility, which have strengthened research into their application in industrial development.

The potential of lipid-based nanocarriers to enhance the sex reversal efficacy of MT in fish has not been investigated. Designing a lipid-based nanodelivery system containing 17 alpha-methyltestosterone (MT) could offer a promising masculinization platform to induce single-sex red tilapia (*Oreochromis* spp.) cultures. Therefore, the aim of the present study was to optimize a ready-to-use masculinization nanoparticle that could be delivered orally to red tilapia fry to reduce the dose and duration of MT administered orally to fish. We characterized four classes of lipid-based nanoparticles, including solid lipid nanoparticles (SLN) and nanostructured lipid carriers (NLC) with and without the integration of alkyl polyglucoside (APG), from which we selected a promising nanocarrier platform to use as an alternative for the oral delivery of MT in a sex reversal study in red tilapia fry.

## 2. Materials and Methods

### 2.1. Chemicals

The hormone 17 alpha-methyltestosterone was purchased from Sigma (Bangkok, Thailand). Medium-chain triglycerides (MCT) and ethoxydiglycol were purchased from PC Intertrade (Bangkok, Thailand). Span 80, Tween 20, glycerol and Synperonic PE/F68 were purchased from Croda (Bangkok, Thailand). Montanov 82 was purchased from Chemico (Bangkok, Thailand). All chemicals, materials and water were purified and of laboratory grade. Alkyl-poly glucosides (APG) were purchased from Croda (Bangkok, Thailand).

### 2.2. Formulation and Fabrication of Lipid-Based Nanoparticles

The NLC and SLN, the subtypes of the lipid-based nanoparticles, were fabricated using hot high-energy homogenization techniques [21]. As shown in Table 1, the oil phase was prepared using 17 alpha-methyltestosterone as a solid lipid (200 mg) and MCT as a liquid lipid (5 g only for the fabrication of NLC).

The oil mixture was dissolved and mixed with the oil phase surfactant by stirring at 200 rpm and 70 °C on a hotplate. Ethoxydiglycol (3 g), cetearyl alcohol and cocoglucoside (1 g), and Span 80 (3 g) were added to the oil mixture and stirred until completely dissolved. An aqueous phase mixture was prepared by mixing purified water, Tween 20 (3 g), glycerol (2.5 g) and Synperonic PE/F68 (2 g) on a hotplate stirrer, and the mixture was poured into the lipid phase. A high-speed homogenizer (T25, IKA, Wilmington, NC, USA) at 10,000 rpm was used to prepare the pre-emulsion for 5 min. Furthermore, the mixtures were sonicated in a sonicator unit (Qsonica sonicators, Newtown, CT, USA) at a 20 amp pulse of 30 s on and off at 5 s intervals for 5 min to prepare the testosterone–nanostructure lipid nanocarriers (MT-NLC). Testosterone–solid lipid carriers (MT-SLN) were also fabricated by following the same preparation steps as for NLC by without adding MCT (the liquid lipid). The MT-NLC and MT-SLN were cooled to an ambient temperature and used for subsequent studies. For the APG-integrated lipid-based nanoparticles, APG was added to a lipid mixture to fabricate alkyl-polyglucoside-integrated nanostructured lipid carriers (MT-APG-NLC) and alkyl-polyglucoside-integrated solid lipid nanoparticles (MT-APG-SLN).

### 2.3. Characterization of the Lipid-Based Nanoparticles

#### 2.3.1. Dynamic Light Scattering

The hydrodynamic diameter, particle distribution and particle surface charge of lipid-based nanoparticles were characterized by dynamic light scattering (DLS) using a zetasizer (Nano ZS, Malvern Instrument, Malvern, Worcs, UK). DLS experiments were performed using a He-Ne laser (λ0 = 633 nm, θ = 173°). Before the measurements, the samples were diluted 50 times in purified water. All reported data were from experiments performed in triplicate at 25 °C. The particles’ stability was detected using changes in the particles’ size and the particles’ phase separation after incubation under various storage conditions (4, 25 and 45 °C) for two months.

#### 2.3.2. Transmission Electron Microscopy

The particles’ size and morphology were characterized using a transmission electron microscope (TEM; JEOL-2100 Plus, JEOL, Akishima, TYO, Japan). Samples were diluted 50-fold in deionized water, dropped onto a carbon grid, stained with phosphotungsten (1% *w*/*w*), dried in a dry cabinet overnight and observed under 100 kV with a magnification of ×25k and ×100k. The particles’ size and morphology were determined for an average of more than 50 particles for each sample.

#### 2.3.3. Encapsulation Efficiency

The encapsulation efficiency (EE) was evaluated using an Amicon centrifuge filter (30 kDa MWCO, cellulose membrane, Millipore, Burlington, MA, USA). The lipid-based nanoparticles (10 mL) were loaded into the filter membrane and centrifuged (MX-307, Tomy, Tokyo, Japan) by following the manufacturer’s protocol. The supernatant was filtrated through a syringe filter (0.22 µm nylon membrane, Millipore, Burlington, MA, USA), referred to as the “non-encapsulated drug concentration”. Furthermore, the encapsulated lipid particles were extracted using the liquid extraction method and filtrated through a syringe filter (0.22 µm nylon membrane, Millipore, Burlington, MA, USA) to be used as the encapsulated drug concentration [22,23]. High-performance liquid chromatography coupled with a photodiode array detector (HPLC-PAD 2998 separation unit and PDA, Water, Milford, MA, USA) was validated and used with the following conditions: an Atlantis C-18 column (4.6 mm × 250 mm, Waters, Milford, MA, USA), a detection wavelength of 240 nm, a mobile phase of acetonitrile: MeOH (60:40, *v*/*v*) and a flow rate of 1.0 mL/min. The encapsulation efficiency (EE) of testosterone in lipid-based nanoparticles was calculated by using the following equation [24].
EE (%) = [(*C_i_* − *C_f_*)/*C_i_*)] × 100(1)
where *C_i_* corresponds to the initial concentration of testosterone added to LBN particles and *C_f_* represents the concentration of free testosterone.

#### 2.3.4. Release Kinetics of Testosterone in Lipid-Based Nanoparticles

The release profiles of MT from NLC and APG-NLC were determined and compared with those of free testosterone (diluted in ethanol) using a dialysis membrane (3500 MWCO regenerated cellulose tubular membrane, CelluSep, Seguin, TX, USA). The samples were placed in a dialysis membrane and incubated in the released buffer (PBS buffer pH 7.4, 25 °C). The released buffer was collected and replaced with fresh released buffer at regular intervals. The concentrations of testosterone were determined using HPLC-PAD as described above. Moreover, Avrami’s equation was used to determine the release mechanism and release constant of MT-synthesized-LBN.
*R* = exp [−(*kt*)^n^](2)
where *R* corresponds to the retention of MT at time (*t*), and *n* and *k* correspond to the release mechanism and release rate constants, respectively.

### 2.4. Masculinization of Red Tilapia 

#### 2.4.1. Ethical Statement

The protocols for the fish experiments used in this research were approved by the animal ethics committee of Chulalongkorn University Animal Care and Use Committee (CUACUC; approval No. 1931077).

#### 2.4.2. Fish Production, Rearing and Experimental Protocol

The red tilapia fry (8-day-old swim-up (Stage 5)) were obtained from a hatchery unit from a GMP Certified fish farm in Ayutthaya Province, Thailand. Furthermore, the red tilapia swim-up fry were acclimatized (3 days) and randomly divided into 10 treatments (400 fish/treatment). The experimental groups were Control 1 (commercial feed, Charoen Pokphand (CP) 9931, mixed with ethanol (ET) without MT), T_1_ (30 ppm of MT in the feed for 14 days), T_2_ (30 ppm of MT in the feed for 21 days), T_3_ (60 ppm of MT in the feed for 14 days), T_4_ (60 ppm of MT in the feed for 21 days), Control 2 (commercial feed mixed with APG-NLC without MT), T_5_ (30 ppm of APG-NLC loaded with MT in the feed for 14 days), T_6_ (30 ppm of APG-NLC loaded with MT in the feed for 21 days), T_7_ (60 ppm of APG-NLC loaded with MT in the feed for 14 days) and T_8_ (60 ppm of APG-NLC loaded with MT in the feed for 21 days) with four replications. The red tilapia fry were distributed into 100 L fiber tanks containing dechlorinated fresh water under continuous aeration. The water (50%) was exchanged daily, and the tanks were replenished with fresh dechlorinated water. The experiment was performed at a controlled temperature within the range of 25–33 °C for air temperature and 25–28 °C for water temperature. The pH and content of dissolved oxygen (DO) were also determined to be within the range of 7.4–8.2 and 5.0–6.0 mg/L, respectively.

To prepare the MT-ET diet, MT was diluted in ethanol, sprayed, mixed with the commercial diet and air-dried. In addition, the MT-APG-NLC diet was prepared by directly mixing the APG-NLC loaded with MT and a commercial diet. Fish in all groups were fed at 10% of body weight five times per day. The fish were fed with a commercial diet until the 45th day of the trial, after completing the sex reversal treatment for 14 days (T_1_, T_3_, T_5_ and T_7_) and 21 days (Control 1, T_2_, T_4_, Control 2, T_6_ and T_8_). At the end of 21 days of feeding, the fish in each group were counted, and the body weight was recorded to indicate the growth performance of the red tilapia fry.

#### 2.4.3. Length–Weight Relationship

The length–weight relationship (LWR) between the total length and body weight was expressed as [25]
W = *a*L*^b^*(3)
where W is the body weight of the fish (g), L is the total length (cm), *a* is the rate of the change in weight with a constant length and *b* is the weight of one unit of length (slope) estimated from the linear regression equation transformed by taking the natural logarithm (log) of both sides.
Log W = Log *a* + *b*. Log L(4)
when *b* = 3, an isometric pattern of growth occurs but when *b* is >3 or <3, an allometric pattern emerges, which may be positive when the length increases relative to the body’s thickness or negative when the length increases relative to the body’s thinness [26].

#### 2.4.4. Relative Condition Factor

The relative condition factor (K_n_) was developed for assessing the health condition of tilapia under all the treatments.
K_n_ = W_o_/W_c_(5)
where W_o_ is the observed weight and W_c_ is the calculated weight [27]. Good growth condition of the fish was when K_n_ > 1, while the organism was in poor growth condition compared with an average individual of the same length when K_n_ < 1.

#### 2.4.5. Sex Reversal Ratio Determined by the Gonad Squashing Technique

After 45-days of the experimental period, all the fish in each group were collected and anesthetized using clove oil for sex determination. The gonads of the fish were collected using fine forceps and placed on glass slides. The samples were stained with an aceto-carmine staining solution, squashed with cover slides and examined under a light microscope [28].

### 2.5. Statistical Analysis

The results were expressed as the means ± SD, and the statistical analysis was conducted by using IBM SPSS statistics software (SPSS Inc., Version 28, Chicago, IL, USA). The effects of the treatments on different parameters were analyzed by one-way analysis of variance (ANOVA), followed by Tukey’s post hoc tests for multiple comparisons. The effect of dietary supplementation with MT and MT-APG-NLC on the sex ratio (0 = male; 1 = female) of red tilapia fry was analyzed using a binary logistic regression model. Linear regression analysis was used to assess the length–weight relationship. A value of *p* < 0.05 was considered statistically significant.

## 3. Results

### 3.1. Characterization of the Lipid-Based Nanoparticles

The size of all four synthesized nanoparticles was lower than 125 nm. APG-SLN (122.23 nm) and APG-NLC (84.21 nm) were both smaller than bare SLN (125.30 nm) and NLC (94.56 nm). The average surface charge and polydispersity index (PDI) of the synthesized nanoparticles were lower than −20 mV and 0.3, respectively, as shown in Table 2 and Figure 1A,B. The stability of the synthesized nanoparticles was observed over time. After 2 months, solid sediments accumulated in the SLN formulation, while the bare NLC and APG-NLC provided higher particle stability without phase separation or solid sedimentation (data not shown). Importantly, APG-NLC had smaller particle sizes immediately after particle synthesis compared with the bare NLC. In terms of the particles’ stability, APG-NLC showed no significant change in the size of particles after being stored under cool (4 °C) and ambient (25 °C) conditions for 2 months.

However, under accelerated (45 °C) conditions, the size of the particles tended to increase to 144.83 nm without phase separation and sedimentation. For the bare NLC particles, the size was increased, and phase separation was observed under accelerated (45 °C) conditions. In addition, TEM images showed the spherical particles of LBNs within the range of 90–120 nm (Figure 2A–D).

### 3.2. Encapsulation Efficiency and Study of the In Vitro Release of MT from LBN

Chemical characterization using HPLC demonstrated that the encapsulation efficiency (EE) values of the synthesized lipid-based nanoparticles were 70.52 ± 3.42%, 70.06 ± 2.24%, 83.91 ± 4.01% and 84.75 ± 13.56% for MT-SLN, MT-APG-SLN, MT-NLC and MT-APG-NLC, respectively. On the basis of the stability and encapsulation efficiency of the synthesized particles, MT-NLC, MT-APG-NLC and MT-ET were selected for evaluation in an in vitro release study. During the first three hours, MT released from MT-NLC, MT-APG-NLC and free MT (dissolved in EtOH) showed release percentages of 16.9 ± 1.0%, 14.1 ± 2.0% and 11.6 ± 2.4%, respectively. A gradual release of MT from MT-NLC, MT-APG-NLC and MT-ET was observed, which reached 38.6 ± 2.5, 30.9 ± 4.1 and 11.4 ± 1.8% within 8 h and extended to the maximum in 24 h (Figure 3A).

After 24 h, there was no further release of MT into the receiving buffer, which may be due to MT’s insoluble nature in water, leading to sedimentation in the aqueous media and the inability to diffuse or pass through the dialysis membrane. The release kinetics of the MT loaded in the synthesized LBN applied Avrami’s equation to determine the release mechanism (*n*) and release rate constant (*k*) (Table 3 and Figure 3B).

The release mechanisms (*n*) were found to be close to 1, being 0.95 for MT-NLC and 1.12 for MT-APG-NLC. Simultaneously, the release mechanism (*n*) of MT-ET was classified in two separate steps: 0.36 for the first two hours and 0.04 after three hours. Moreover, the release rate constant of MT from MT-NLC and MT-APG-NLC was found to be 0.587 × 10^−1^/h and 0.637 × 10^−1^/h, respectively. The release constant of MT-ET was lower than 0.01 × 10^−1^/h, as shown in Table 3 and Figure 3B. Importantly, further analysis was carried out by using APG-NLC based on the particles’ size and stability, the encapsulation efficiency and the results of the release kinetics.

### 3.3. Red Tilapia Masculinization Using Testosterone Loaded in Lipid-Based Nanoparticles

The sex of the treated red tilapia fry was identified by staining the gonads with acetocarmine and determined on the basis of the morphological characteristics. For the gonad squashing technique, the male gonad was identified by its smooth tubular shape and the absence of an oocyte (Figure 4A), while the female gonad was identified by its round-shaped oocytes (Figure 4C). The gonadal tissue was also confirmed by histological observations, in which the male gonad was identified by the appearance of spermatogonia (Figure 4B) and the female gonadal tissue was characterized by the round shape of the oocyte (Figure 4D).

ANOVA showed a significant interaction (*p* < 0.01) between the effects of the concentrations of MT and MT-APG-NLC and the duration of the sex reversal period on the male ratio of red tilapia. The logistic regression model found that the male percentage in T_1_ (*b* = 0.260, *p =* 0.017), T_2_ (*b* = −0.284, *p =* 0.009), T_4_ (*b* = −2.165, *p =* 0.000), T_5_ (*b* = −1.297, *p =* 0.000), T_6_ (*b* = −4.205, *p =* 0.000) and T_8_ (*b* = −0.698, *p =* 0.000) was significantly higher than that in Control 1, Control 2, T_3_ and T_7_ (Table 4).

The T_6_ treatment displayed the highest male percentage (98.5%), which was significantly higher (*p* < 0.05) than that of all the other experimental groups. The chi-square analysis of Control 1 (Wald *X*^2^ = 0.012, *p =* 0.914) and Control 2 (Wald *X*^2^ = 2.301, *p =* 0.129) indicated that males and females were equally represented. Likewise, there was no significant change in the normal 1:1 sex ratio in fish fed with T_3_ (Wald *X*^2^ = 3.389, *p =* 0.066) and T_7_ (Wald *X*^2^ = 0.047, *p =* 0.828). All of the studied fish were found to be either male or female.

The survival rate was more than 85% in the MT and MT-APG-NLC groups, which was not significantly decreased (*p* > 0.05) compared with the control groups (Table 4). The production cost (in Thai baht) of the sex reversal treatments of red tilapia was estimated from the cost of MT, LBN and ethanol used in this experiment, based on equal amounts of the diets and the fish. The production cost was reduced by 32.90% (Table 4) under the T_6_ treatment compared with the conventional treatment (T_4_). Furthermore, the final body weight of red tilapia was also measured in the group that received the sex reversal diets for 21 days. The final body weights of red tilapia were similar among the controls, MT-APG-NLC (30 ppm), MT-APG-NLC (60 ppm), MT (30 ppm) and MT (60 ppm) groups (Figure 5). There was a significant decrease in the body weight of red tilapia fed MT-APG-NLC (60 ppm) compared with the other treatment groups.

The length–weight relationship was used to determine the growth performance of hormone-treated tilapia (Table 5). The calculated *b*-values were lower than 3.0 for all treatment groups, which indicated negatively allomeric growth behavior and implied that the length of tilapia increased relative to body thickness. The higher r^2^ values (>0.95) obtained in the assessment of LWR under all the treatments suggested a good quality of fit for the predictions of the linear regression. Furthermore, a significant correlation (*p* < 0.05) was observed between the length and weight under all the treatments. The relative condition factor (K_n_) was higher than 1.0, demonstrating that the fish were in good condition during the treatment period.

## 4. Discussion

Methyl testosterone is unlikely to have a detrimental effect on the flesh of fish after the sex reversal treatment (SRT) has been discontinued; nonetheless, its effects on the major organs of fish, the metabolic profiles and nucleic acids are inconclusive [29]. However, fish treated with synthetic MT had altered enzymatic metabolic pathways [30], as well as a substantial increase in plasma creatine phosphokinase activity and DNA fragmentation in the liver tissue [31]. The dynamics of the ecosystem may be affected if tilapia hatcheries discharge steroids into the aquatic environment [32]. Furthermore, in certain natural populations, this may result in more males than females, leading to long-term reproductive failure [33]. It has been shown that the accumulation of MT in pond sediments and its detection in the soil occurred approximately 3 months after SRT. Importantly, many countries have limited the administration of synthetic hormones and antibiotics in fish farms because of chemical residues and antimicrobial resistance [34]. Therefore, the application of a potential nanocarrier platform as an alternative for the dietary administration of MT for reversing the sex of tilapia species is highly demanded.

Lipid-based nanocarriers have been shown to be a landmark in drug and hormone delivery. The application of LBN loaded with steroid derivatives, such as exemestane, spironolactone or methylprednisolone aceponate, has been developed and approved to enhance their in vivo bioavailability [35,36,37]. Nanoparticles have been integrated into hormone delivery applications such as luteinizing hormone-releasing hormone (LHRH) encapsulated in chitosan–gold conjugated nanoparticles to induce the reproduction efficiency of *Cyprinus carpio* [38]. In our study, we investigated the efficacy of lipid nanocarriers containing 17 MT for dietary administration as a masculinization agent to enhance the all-male ratio in red tilapia. The average surface charge was lower than −20 mV, and the polydispersity index (PDI) of 0.3 indicated that the MT-SLN, MT-APG-SLN, MT-NLC and MT-APG-NLC were assuredly stable [39]. APG has been previously used as a stabilizer for solid lipid nanoparticles for transdermal and cosmetic applications, which can reduce the particles’ size and induce the physical stability of the nanoparticles [40,41]. In this study, APG also acted as a surfactant in the formulation; therefore, with increasing concentrations, the size of APG nanoparticles decreased [42]. The solid sediments that appeared in the SLN formulation might be attributed to the recrystallization property of MT in the absence of an oil carrier (data not shown) [43]. NLC particles were formulated with an appropriate ratio of the liquid lipid and the solid lipid; it formed an amorphous structure that prevented the expulsion of testosterone to the extracellular matrix or the environment [44,45,46]. In contrast, the crystalline structure of SLN could induce the expulsion and sedimentation of the drug during storage [45,47,48]. The size of the synthesized APG-NLC particles was smaller than that of bare NLC particles and provided greater thermal stability under all storage conditions. This effect can be explained by the fact that APG can act as a non-ionic and biodegradable surfactant [49].

Furthermore, an in vitro investigation showed that the amount of MT released, which was dissolved in ethanol from 3 h to 24 h, could be classified into two separate steps. In both steps, the values of the release mechanism were lower than 1, which suggested that the MT was released through a diffusion mechanism, while the MT loaded in lipid-based nanoparticles was released by first-order release kinetics [50]. The rates of MT released from MT-NLC and MT-APG-NLC were constantly higher than those of MT-ET. The results showed that the release of MT was better in aqueous matrices, which confirmed the benefits of lipid-based nanoparticles as encapsulation agents and for prolonging the release of the androgen hormone.

The conventional sex reversal treatment (MT) has the potential to disrupt the natural differentiation process, even after its initiation, by interfering with normal embryonic gene expression patterns and physiological regulation, resulting in a skewed sex ratio towards males [51]. Moreover, the androgen-induced masculinization of tilapia gonads is correlated with the upregulated expression of *dmrt1* [52,53,54]. In the present study, the T_4_ treatment produced 89.7% sex-reversed males in red tilapia. Our finding is in contrast with previous studies that reported 100% males in commercial tilapia farms [55,56], which was possibly due to the various nursing techniques and climates. Importantly, red tilapia fry in the T_6_ group produced 98.5% males, which is a significantly higher percentage of male tilapia compared with previous studies that found less than 70% sex-reversed males after the administration of 30 mg/kg of conventional MT for 28 days [6,57]. Our findings also confirmed that lipid nanoparticles could be a promising delivery system for androgen hormones. Furthermore, the current findings were corroborated by the in vitro release results, which revealed that APG-NLC could induce drug solubility and controlled drug release in aqueous media. This indicated that MT-APG-NLC (30 ppm) has the potential to be an alternative for delivering sex-reversal hormones to farmed tilapia. We postulated that APG-NLC increased the solubility of MT, which activates the expression of *dmrt1*, an important mechanism underpinning gonadal masculinization in red tilapia that requires further investigation. The higher dose of MT-APG-NLC (60 ppm) resulted in a significant decrease in the percentage of males and their growth, which may be due to a reduction in the intake of feed. Moreover, the reduction in the percentage of male tilapia as the concentration of MT-APG-NLC increased could possibly be due to paradoxical feminization and the aromatization of MT to estradiol, which expresses the female sex hormone [4,58,59]. These results showed that a lower dose (30 ppm) of MT-APG-NLC was more effective than the conventional sex reversal treatment currently used in red tilapia.

The conventional sex reversal treatment of tilapia results in a varied survival rate of fish [60,61,62,63]. Factors that affect the survival of fish include the hormones (natural or synthetic), dosage, feeding frequency and the duration of the sex reversal treatment [58]. All sex reversal treatments achieved more than 80% survival, which is consistent with the 80% survival rate obtained in seed production tilapia farms [56]. Furthermore, of the groups examined here, T_6_ had the lowest production costs, indicating that the MT-APG-NLC treatment is a more cost-effective alternative to the conventional sex reversal treatment for producing all-male red tilapia fry.

The negative allometric growth showed that the LWRs of the control group were in good agreement with those of the MT treatment group. Similarly, Nile tilapia and red tilapia showed negative allometric growth behavior [64,65]. Importantly, a relative condition factor of >1 was found for all the treatments, suggesting good fish health, better feed intake and proper environmental conditions. The results are in accordance with previous studies [65,66].

## 5. Conclusions

APG-integrated lipid-based nanoparticles are a new LBN platform for delivering the androgen hormone MT as a sex reversal agent to tilapia via dietary administration. APG-NLC achieved high encapsulation efficiency, thermal stability and cost-effectiveness for the masculinization of tilapia with MT when delivered orally at 30 ppm for 21 days. This platform is a promising way to reduce the dose of artificial sex reversal hormones used in red tilapia aquaculture for masculinization.

## Figures and Tables

**Figure 1 animals-13-01364-f001:**
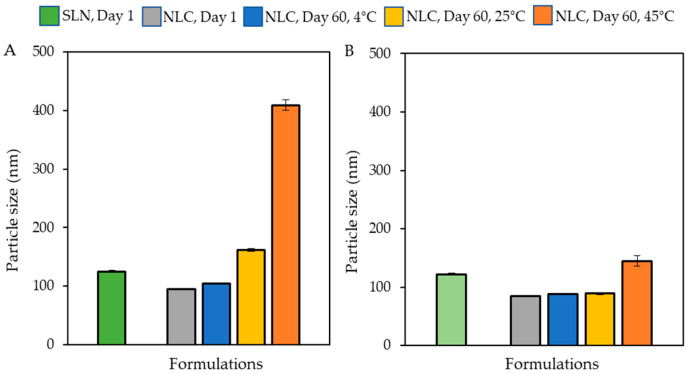
Size and stability of unmodified lipid-based nanoparticles, including solid lipid nanoparticle (SLN), nanostructured lipid carriers (NLC) (**A**) and alkyl polyglucoside integrated with solid lipid nanoparticles (APG-SLN) and alkyl polyglucoside integrated with nanostructured lipid carriers (APG-NLC). (**B**) The samples were stored at 4, 25 and 45 °C for 2 months.

**Figure 2 animals-13-01364-f002:**
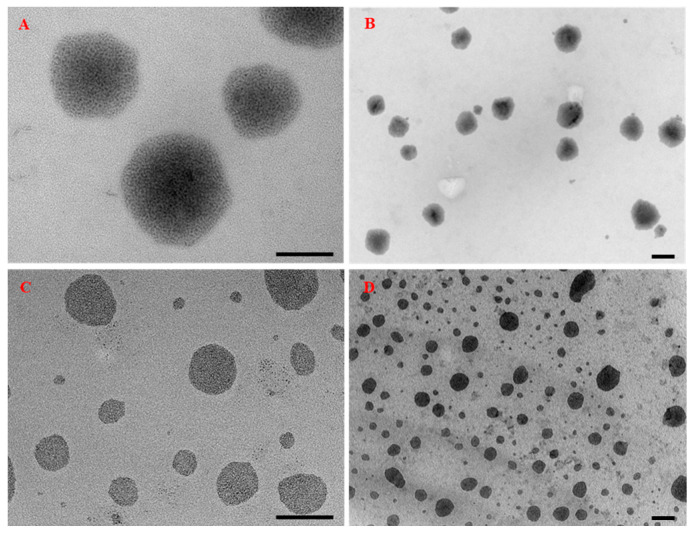
Morphological observations of the nanostructured lipid carriers (NLC) (**A**,**B**) and alkyl polyglucoside integrated with nanostructured lipid carriers (APG-NLC) (**C**,**D**) loaded with 17 alpha-methyltestosterone using transmission electron microscopy (TEM). Scale bar: 100 nm.

**Figure 3 animals-13-01364-f003:**
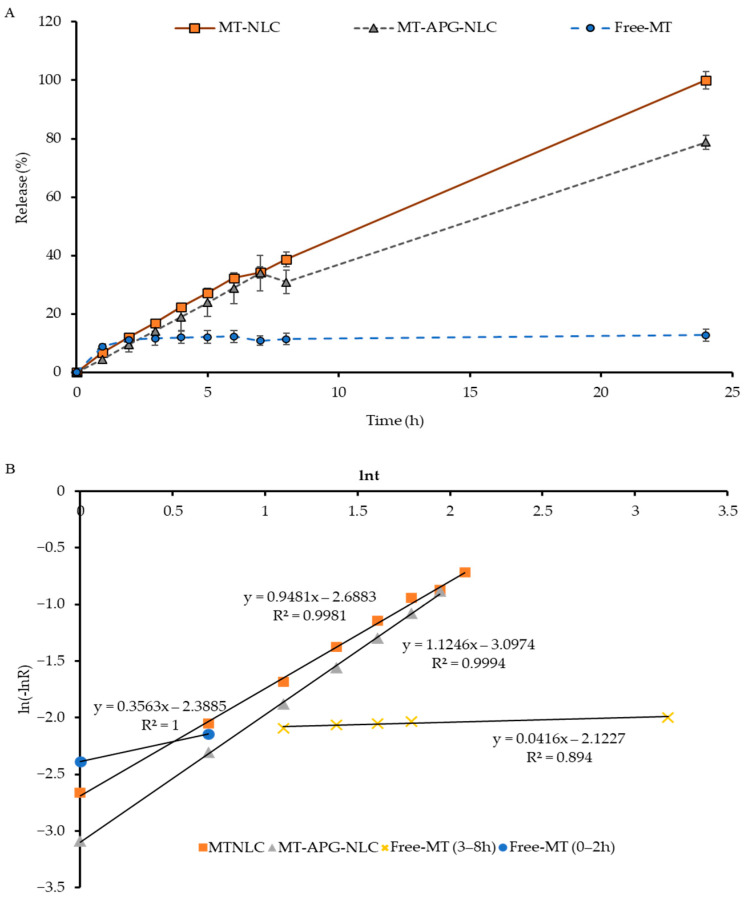
In vitro cumulative release profile of MT from the synthesized LBNs with and without a polyglucoside structure and MT-ET (non-encapsulated MT) using the HPLC-PAD technique (*n* = 3 for each group) (**A**) and an evaluation of the release kinetics of MT-NLC, MT-APG-NLC and MT-ET using Avrami’s equation (**B**).

**Figure 4 animals-13-01364-f004:**
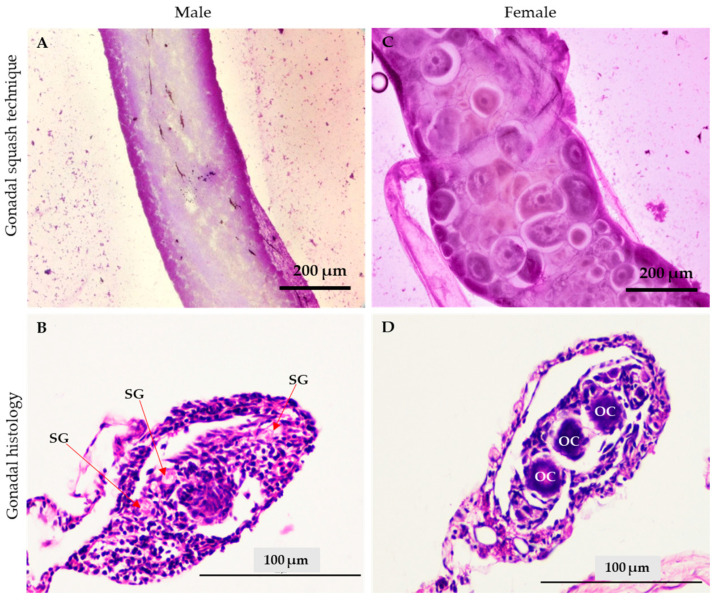
Gonadal morphology of the red tilapia ovary and testis using the gonad squashing technique (**A**,**C**) and histological observations (**B**,**D**) at 45 dph after feeding with MT-APG-NLC (60 ppm) for 21 days. SG: spermatogonia; OC: oocytes.

**Figure 5 animals-13-01364-f005:**
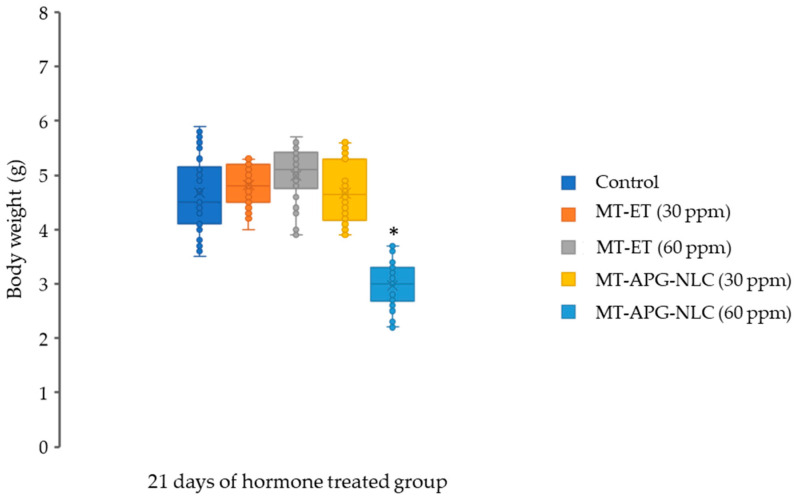
Box plot of the body weight of red tilapia after being fed with the control diet (hormone-free diet), MT-ET (30 and 60 ppm) and MT-APG-NLC (30 and 60 ppm) for 21 days and then continuously fed with control diet until 45 dph. * *p* < 0.05 indicates significant differences among the different treatment groups.

**Table 1 animals-13-01364-t001:** Composition (percent by weight) of four different formulations of MT-LBNs.

	Formulations			
Composition	SLN	APG-SLN	NLC	APG-NLC
17 α-methyltestosterone	0.2	0.2	0.2	0.2
Medium chain triglyceride (MCT)	0.0	0.0	5.0	5.0
Alkyl-polyglucoside	0.0	2.0	0.0	2.0
Span 80	3.0	3.0	3.0	3.0
Ethoxydiglycol	4.0	4.0	3.0	3.0
Tween 20	3.0	3.0	3.0	3.0
Synperonic PE/F68	2.0	2.0	2.0	2.0
Glycerol	2.5	2.5	2.5	2.5
DI water	85.3	83.3	81.3	79.3
Total	100.0	100.0	100.0	100.0

SLN: solid lipid nanoparticles; NLC: nanostructured lipid carriers; APG: alkyl polyglucoside.

**Table 2 animals-13-01364-t002:** Physical properties of LBNs using the dynamic light scattering technique after loading with 17-alpha methyltestosterone.

Formulation	Particle Size (nm)	Surface Charge (mV)	Polydispersity Index
SLN	125.30 ± 1.34	−30.4 ± 2.43	0.211 ± 0.002
APG-SLN	122.23 ± 1.50	−34.0 ± 1.47	0.228 ± 0.020
NLC	94.56 ± 0.17	−28.5 ± 3.50	0.148 ± 0.010
APG-NLC	84.21 ± 0.60	−30.9 ± 3.48	0.123 ± 0.010
MT-ET	ND	ND	ND

Values are the mean ± SD of triplicate samples. ND: not detected; SLN: solid lipid nanoparticles; APG-SLN: solid lipid nanoparticles of alkyl polyglucoside; NLC: nanostructured lipid carriers; MT-ET: 17 alpha-methyltestosterone-ethanol.

**Table 3 animals-13-01364-t003:** Parameters of the release kinetics (release rate constant (*k*), release mechanism (*n*) and *R*^2^) of MT from the encapsulation particles after fitting the release profile with Avrami’s equation.

Sample	*k* (10^−1^/h)	*n*	*R* ^2^
MT-ET (0–2 h)	0.012	0.3563	0.9521
MT-ET (3–8 h)	0.000	0.0416	0.894
MT-NLC	0.587	0.9481	0.9981
MT-APG-NLC	0.637	1.1246	0.994

Values are the mean ± SD of triplicate samples. MT-ET: 17 alpha-methyltestosterone–ethanol; MT-NLC: 17 alpha-methyltestosterone– nanostructured lipid carriers; MT-APG-NLC: 17 alpha-methyltestosterone–alkyl polyglucoside–nanostructured lipid carriers.

**Table 4 animals-13-01364-t004:** Proportion of males, survival rate and cost of production of red tilapia fry fed with MT and MT-APG-NLC for 14 d and 21 d, followed by a commercial diet for 45 d.

Treatments	Male (%)	No. of Fish	Male: Female (1:1)	*p*-Value	*b* ± SE	Exponentiation of *b*	Wald *X*^2^	Survival Rate (%)	Cost (THB/Fish)	Compared with T_4_ (%)
Control 1	50.3 ± 3.2 ^ab^	340	1:0.99	0.914	−0.012 ± 0.108	0.988	0.012	91.5 ± 2.1	-	-
T_1_	43.5 ± 3.8 ^a^	340	1:1.29	0.017	0.260 ± 0.109	1.297	5.662	91.0 ± 4.1	0.0011	−80.49
T_2_	57.1 ± 4.5 ^b^	340	1:0.76	0.009	−0.284 ± 0.110	0.753	6.731	89.5 ± 2.9	0.0029	−50.00
T_3_	55.0 ± 2.1 ^b^	340	1:0.82	0.066	−0.201 ± 0.109	0.818	3.389	88.8 ± 3.9	0.0023	−60.98
T_4_	89.7 ± 3.6 ^c^	340	1:0.11	0.000	−2.165 ± 0.178	0.115	147.160	88.5 ± 3.7	0.0059	0.00
Control 2	45.9 ± 2.0 ^a^	340	1:1.19	0.129	0.165 ± 0.109	1.179	2.301	90.3 ± 3.7	-	-
T_5_	78.5 ± 4.8 ^c^	340	1:0.27	0.000	−1.297 ± 0.132	0.273	96.404	91.3 ± 2.1	0.0015	−73.85
T_6_	98.5 ± 1.4 ^d^	340	1:0.02	0.000	−4.205 ± 0.451	0.015	87.097	92.0 ± 2.9	0.0039	−32.98
T_7_	50.6 ± 2.8 ^a^	340	1:0.98	0.828	−0.024 ± 0.108	0.977	0.047	87.3 ± 4.4	0.0031	−47.70
T_8_	66.8 ± 1.7 ^b^	340	1:0.50	0.000	−0.698 ± 0.115	0.498	36.711	85.5 ± 1.7	0.0079	+34.03

The results are expressed as the mean ± SD. Different superscript letters in a column indicate significant differences (*p* < 0.05) from the control. Chi-square values with *p* < 0.05 indicate a significant deviation from the expected sex ratio of 1:1. *b* = regression coefficient; *p*-value: probability of a male occurring; *X*^2^: chi-square values.

**Table 5 animals-13-01364-t005:** Length–weight relationship of red tilapia fry fed with MT and MT-APG-NLC for 21 days, followed by a commercial diet for 45 days.

Treatments	Control	MT-ET (30 ppm)	MT-ET (60 ppm)	MT-APG-NLC (30 ppm)	MT-APG-NLC (60 ppm)
N	50	50	50	50	50
L_min-max_ (cm)	2.2–8.8	2.9–7.9	2.9–8.7	4.3–8.3	2.0–7.0
W_min-max_ (g)	1–7.3	1.9–6.8	1.6–7.4	2.9–6.5	1.1–5.2
*a*	−0.372	−0.258	−0.351	−0.307	−0.363
*b*	1.3423	1.196	1.286	1.225	1.289
SE (*b*)	0.031	0.038	0.028	0.018	0.020
CI (*b*)	1.280–1.405	1.119–1.273	1.230–1.341	1.189–1.260	1.249–1.330
r^2^	0.975	0.953	0.978	0.990	0.988
*p*	0.000	0.000	0.000	0.000	0.000
*t*-test sig	0.000	0.000	0.000	0.000	0.000
Growth behavior	Negative allometry	Negative allometry	Negative allometry	Negative allometry	Negative allometry
K_n_	1.003	1.002	1.001	1.000	1.001
Min-Max	0.818–1.445	0.830–1.148	0.876–1.102	0.967–1.067	0.926–1.072
SE	0.011	0.009	0.006	0.003	0.005

N: sample size; L: length (cm); W: weight (g); Min: minimum; Max: maximum; a: intercept; b: slope of the equation; SE: standard error; CI (b): confidence intervals of b; r^2^: coefficient of determination; *p*: significance of the regression, with *p* significant at <0.05; K_n_: relative condition factors. Significance was determined by the *t*-test to verify if *b* was significantly different from the consensus (*b* = 3). The growth behavior was based on *b*.

## Data Availability

The data presented in this study are available on request from the corresponding author.

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
