# Peer review of "Masculinization of Red Tilapia (Oreochromis spp.) Using 17α-Methyltestosterone-Loaded Alkyl Polyglucosides Integrated into Nanostructured Lipid Carriers"

_animals, 2023, doi:10.3390/ani13081364_

Round 1

Author Response

Response to Reviewer 1 Comments

In the article entitled “Masculinization of Red Tilapia (Oreochromis spp.) Using 17α-Methyltestosterone-Loaded Alkyl Polyglucosides Integrated into Nanostructured Lipid Carriers”, the authors investigated the effects of 17α-Methyltestosterone on masculinization of red tilapia. Author demonstrated that nanotechnology-based drug delivery systems represent a novel and promising strategy for improving oral absorption of these androgen hormones, resulting in a 50% w/w hormonal dosage reduction during the sex reversal period. Their idea and data might shed light. However the manuscript can be revised in some points, as reported here:

Point 1: In the MATERIALS and METHODS Section: Please absolutely add Ethics statement to the text.

Response: Thank you for the suggestion. We have included an ethics statement in the materials and methods section.

In the revised manuscript:

Lines 181–184:

Ethics statement

The protocols for the fish experiments used in this research were approved by the animal ethics committee of Chulalongkorn University Animal Care and Use Committee (CUACUC; Approval No. 1931077).

Point 2: Why 8-day-old fry?

Response: Thank you for the observation. The red tilapia fry (8-day-old swim-up (stage 5)) were obtained from a "Hatchery unit from a GMP Certified fish farm in Ayutthaya province, Thailand”. After collection, red tilapia’s fry were acclimatized for 3 days in the experimental system, i.e., 11 days post-hatching. It is essential to start MT feeding at this age because sex determination takes place 12–15 days post-hatching (Bhujel, 2014).

In the revised manuscript:

Lines 186-189: The red tilapia fry (8-day-old swim-up (stage 5)) were obtained from a "Hatchery unit from a GMP Certified fish farm in Ayutthaya province, Thailand”. Furthermore, red tilapia’s swim-up fry were acclimatized (3 days) and randomly divided into 10 treatments (400 fish/treatment).

Reference

Bhujel, R. C. (2014). A manual for tilapia business management. CABI, UK, 214pp.

Point 3: LINE 191: “5.0–6.0 mg/L and 7.4–8.2” change to be “ 7.4–8.2 and 5.0–6.0 mg/L”.

Response: Thank you for the suggestion. We have already made the changes on Line 201 in the revised manuscript.

Point 4: In the RESULTS Section: LINE 217 please delete “below”.

Response: Thank you for the suggestion. We have already made the changes by deleting below on Line 242 in the revised manuscript.

Point 5: FIGURE 4 is not clear. “SG” in FIG 4A indicates needs to be reconsidered by the author.

Response: Thank you for the excellent observation and suggestion. We have revised Fig. 4A in the revised manuscript. Male gonads were identified by their smooth tubular shape and absence of oocytes, while female gonads were identified by their round-shaped oocytes (Sulaeman and Fotedar, 2017; Sarker, et al., 2022).

Figure 4. Gonadal morphology using gonadal squash technique (A, C) and histological observation (B and D) at 45 dph of red tilapia after feeding with MT-APG-NLC (60 ppm) for 21-days in tilapia ovary and testis. SG: Spermatogonia; OC: Oocytes.

In the revised manuscript:

Lines 308-313: For gonadal squash technique, the male gonad was identified by its smooth tubular shape and absence of an oocyte (Fig. 4A), while the female gonad was identified by its round-shaped oocytes (Fig. 4C). The gonadal tissue was also confirmed by histological observation, in which the male gonad was identified by the appearance of spermatogonia (Fig. 4B) and the female gonadal tissue was characterized by the round shape of the oocyte (Fig. 4D).

References

Sulaeman and Fotedar, R. (2017). Masculinization of silver perch (Bidyanus bidyanus Mitchell 1838) by dietary supplementation of 17α-methyltestosterone. The Egyptian Journal of Aquatic Research, 43(1), 109-116.

Sarker, B., Das, B., Chakraborty, S., Hossain, M. A., Alam, M. M., Mian, S., & Iqbal, M. M. (2022). Optimization of 17α-methyltestosterone dose to produce quality mono-sex Nile tilapia Oreochromis niloticus. Heliyon, 8(12), e12252.

Point 6: Whether MT remains in the fish?

Response: Thank you for the question. MT is classified as a natural structure of steroid hormone, which is derived from cholesterol that is easily metabolized in tilapia and eliminated within an hour or day through bile or urine secretion (Hoga et al., 2018, and Truscott, 1983). Furthermore, a few more studies also reported that the use of hormones does not result in the accumulation of residues in the tissues of treated fish (Abucay & Mair, 1997; Curtis et al., 1991). Therefore, based on the previous study, we postulated that MT can be immediately metabolized and eliminated from the fish organ.

References

Abucay, J. S.; Mair, G. C. Hormonal sex reversal of tilapias: implications of hormone treatment applications in close water systems. Aquaculture Research, Oxford, v. 28, n. 11, p. 841-845, Nov. 1997.

Curtis, L. R., Diren, F. T., Hurley, M. D., Seim, W. K., & Tubb, R. A. (1991). Disposition and elimination of 17α-methyltestosterone in Nile tilapia (Oreochromis niloticus). Aquaculture, 99(1-2), 193-201.

Hoga, C. A., Almeida, F. L., & Reyes, F. G. (2018). A review on the use of hormones in fish farming: Analytical methods to determine their residues. CyTA-Journal of Food, 16(1), 679-691.

Truscott, B. (1983). Steroid metabolism in fish: II. Testosterone metabolites in the bile of the marine winter flounder Pseudopleuronectes americanus and the freshwater Atlantic Salmon Salmo salar. General and comparative endocrinology, 51(3), 460-470.

Point 7: Whether MT content in muscle should be tested?

Response: Thank you for your question. In the previous study, Johnstone et al. (1983) reported that 99% of MT was eliminated from both visceral tissue and carcasses of juvenile tilapia and rainbow trout within 100 h after withdrawal of the MT diet. Furthermore, Goudie et al. (1986) found 1% (about 5 ng MT/g of tissue) of MT in tilapia tissue after withdrawal from the MT diet for 21 days. Importantly, MT-APG-NLC (30 ppm) is recommended to reduce the dose of MT used for the masculinization of red tilapia farming. We believed that, according to the literature, 30 ppm of MT could be metabolized and eliminated from the tilapia body within a week after the withdrawal of MT treatment. Therefore, we do not suggest testing the MT content in muscle.

References

Goudie, C. A., Shelton, W. L., & Parker, N. C. (1986). Tissue distribution and elimination of radio labelled methyltestosterone fed to sexually undifferentiated blue tilapia. Aquaculture, 58(3-4), 215-226.

Johnstone, R., Macintosh, D. J., & Wright, R. S. (1983). Elimination of orally administered 17α-methyltestosterone by Oreochromis mossambicus (tilapia) and Salmo gairdneri (rainbow trout) juveniles. Aquaculture, 35, 249-257.

Point 8: In Discussion and Conclusion, this manuscript may be interesting but some aspects have to be deeply improved to support authors’ idea.

Response: Thank you for the suggestion. We have improved the discussion in the revised manuscript. However, we would be willing to answer further if the reviewer let us know which aspects need to be improved.

In the revised manuscript

Lines 420-425: Importantly, red tilapia fry fed with T6 group produced 98.5% males, which is significantly higher in male tilapia percentage compared with the previous studies that found less than 70% sex-reversed males after feeding with 30 mg/kg of conventional MT administration for 28 days [6,57]. Our finding also confirmed that lipid nanoparticle could be a promising delivery system for androgen hormone.

Lines 446-451: The negative allometric growth showed that the LWRs of the control group were in good agreement with those of the MT-treatment group. Similarly, Nile tilapia and red tilapia showed negative allometric growth behavior [64,65]. Importantly, >1 relative condition factor was found in all the treatments, suggesting good fish health, better feed intake, and proper environmental conditions. The results are in accordance with previous studies [65,66].

Reviewer 2 Report

The development of methods for the hormonal regulation of sex in tilapia is useful for optimizing the aquaculture of this fish species. The authors managed to create alkyl polyglucoside nanocapsules for more efficient delivery of hormones to the body of fry. This is interesting.

Author Response

Response to Reviewer 2 Comments

The development of methods for the hormonal regulation of sex in tilapia is useful for optimizing the aquaculture of this fish species. The authors managed to create alkyl polyglucoside nanocapsules for more efficient delivery of hormones to the body of fry. This is interesting.

Response: Thank you for the excellent comment. We are greatly appreciative of your time.

Reviewer 3 Report

Masculinisation is an economically important methodology for the creation of all male populations in fish in which males exhibit better growth rates. The described methodology, which is based in nanoparticles, is promising. Although it is well described, the researchers should have tested more biomarkers such as the dmrt3 gene, as they mention in the discussion. At least they should examine length weight relationships instead of just final body weight. Apart from that, it is a well written and well structured manuscript

Some specific comments

Was the aim to develop or to optimise? Since there are already available methodologies for masculinization, I suggest to use the term optimise.

Any other welfare markers apart from body weight? 

Line 217: Either “below” or “lower” should be deleted

Lines 373-380. This part is confusing. How could the authors compare their results that are based on nanomaterials with conventional MT administration? Also, the 89,7% is low? Why do the authors mention “only”? I believe it should be rewritten 

Author Response

Response to Reviewer 3 Comments

Point 1: Masculinisation is an economically important methodology for the creation of all male populations in fish in which males exhibit better growth rates. The described methodology, which is based in nanoparticles, is promising. Although it is well described, the researchers should have tested more biomarkers such as the dmrt3 gene, as they mention in the discussion. At least they should examine length weight relationships instead of just final body weight. Apart from that, it is a well written and well structured manuscript.

Response: Thank you for your suggestion. Gene expression is one of the most important factors affecting gonadal differentiation. However, in our present research, the phenotypic trait of sex reversal tilapia can be affected by both gene expression and environmental factors. Therefore, the sex reversal efficiency of hormonal treatment has been determined using two conventional methods, including the gonadal squash technique and histological determination (Sarker et al., 2022, and El-Greisy and El-Gamal, 2012).

Additionally, based on your comment, length-weight relationships were also determined and reported in the result and discussion.

In the revised manuscript

Lines 210-225:

2.4.3. Length-weight relationship (LWR)

Length-weight relationship (LWR) between the total length and body weight was expressed as (Pauly, 1983);

W = aLb

(3)

Where W is the body weight of fish (g), L is the total length (cm), a is the rate of weight of change with length (constant), and b is the weight of one unit length (slope) estimated from the linear regression equation transformed by taking the natural logarithm (Log) of both sides.

Log W = Log a + b. Log L

(4)

Where b when = 3, an isometric pattern of growth occurs but when b is >3 or <3, an allometric pattern emerges, which may be positive when length increases relative to body thickness or negative when length increases relative to body thinness (Froese, 2006).

2.4.4. Relative condition factor (Kn)

Relative condition factor (Kn) was developed for assessing the health condition of tilapia in all the treatments.

Kn = Wo/Wc

(5)

where Wo is the observed weight, and Wc is the calculated weight (Le Cren, 1951). Good growth condition of the fish is deduced when Kn >1, while the organism is in poor growth condition compared to an average individual with the same length when Kn < 1.

Lines 351-365:

Table 5. Length-weight relationship of red tilapia fry fed with MT and MT-APG-NLC for 21 d followed by a commercial diet for 45 d.

 Treatments

Control

MT-ET

(30 ppm)

MT-ET

(60 ppm)

MT-APG-NLC

(30 ppm)

MT-APG-NLC

(60 ppm)

N

50

50

50

50

50

Lmin-max (cm)

2.2–8.8

2.9–7.9

2.9–8.7

4.3–8.3

2.0–7.0

Wmin-max (g)

1–7.3

1.9–6.8

1.6–7.4

2.9–6.5

1.1–5.2

a

-0.372

-0.258

-0.351

-0.307

-0.363

b

1.3423

1.196

1.286

1.225

1.289

SE (b)

0.031

0.038

0.028

0.018

0.020

CI (b)

1.280–1.405

1.119–1.273

1.230–1.341

1.189–1.260

1.249–1.330

r2

0.975

0.953

0.978

0.990

0.988

P

0.000

0.000

0.000

0.000

0.000

t-test sig

0.000

0.000

0.000

0.000

0.000

Growth behavior

Negative

allometry

Negative

allometry

Negative

allometry

Negative

allometry

Negative allometry

Kn

1.003

1.002

1.001

1.000

1.001

Min-Max

0.818–1.445

0.830–1.148

0.876–1.102

0.967–1.067

0.926–1.072

SE

0.011

0.009

0.006

0.003

0.005

N: sample size; L: Length (cm); W: weight (g); Min: Minimum; Max: Maximum; a: intercept; b: slope of the equation; SE: Standard error; CI (b): confidence intervals of b; r2: coefficient of determination; P: significance of regression with P significant at < 0.05; Kn: relative condition factors. t-test significance was conducted to verify if b is significantly different from the consensus b = 3; The growth behavior was deduced based on b.

The length-weight relationship was used to determine the growth performance of hormone-treated tilapia (Table 5). The calculated b-values are lower than 3.0 for all treatment groups, which indicates negatively allomeric growth behavior and implies that tilapia have a relatively slow growth rate and tend to be thinner. The higher r2 values (> 0.95), obtained in the assessment of LWRs in all the treatments, suggest good quality in the prediction of a linear regression. Furthermore, a significant correlation (p < 0.05) was observed in the length and weight of all the treatments. The relative condition factor (Kn) is higher than 1.0, demonstrating that the fish were in good condition during the treatment period.

Lines 446-451:

The negative allometric growth showed that the LWRs of the control group were in good agreement with those of the MT-treatment group. Similarly, Nile tilapia and red tilapia showed negative allometric growth behavior (Malik et al., 2017; Cishahayo et al., 2022). Importantly, >1 relative condition factor was found in all the treatments, suggesting good fish health, better feed intake, and proper environmental conditions. The results are in accordance with previous studies (Cishahayo et al., 2022; Komba et al., 2020).

References

Cishahayo, L.; Yongo, E.; Mutethya, E.; Waithaka, E.; Ndayishimiye, R. Length–weight relationship, condition factor, sex ratio and size at first maturity of the blue‐spotted tilapia (Oreochromis leucostictus) in Lake Naivasha, Kenya. Lakes & Reservoirs: Research & Management 2022, 27, e12417.

El-Greisy, Z. A., & El-Gamal, A. E. (2012). Monosex production of tilapia, Oreochromis niloticus using different doses of 17α-methyltestosterone with respect to the degree of sex stability after one year of treatment. The Egyptian Journal of Aquatic Research, 38(1), 59-66.

Froese, R. Cube law, condition factor and weight–length relationships: history, meta‐analysis and recommendations. Journal of Applied Ichthyology 2006, 22, 241-253.

Komba, E.A.; Munubi, R.N.; Chenyambuga, S.W. Comparison of body length-weight relationship and condition factor for Nile tilapia (Oreochromis niloticus) cultured in two different climatic conditions in Tanzania. International Journal of Fisheries and Aquatic Studies 2020, 8, 44-48.

Le Cren, E.D. The length-weight relationship and seasonal cycle in gonad weight and condition in the perch (Perca fluviatilis). The Journal of Animal Ecology 1951, 201-219.

Malik, A.; Abbas, G.; Soomro, M.; Shah, S.; Asadullah, A.; Bhutto, A.; Jamali, G.; Roonjho, Z. Length-weight relationship and condition factor of red tilapia (Hybrid) reared in cemented tanks of Sun-bright Red Tilapia and ornamental hatchery-Karachi, Sindh-Pakistan. Sindh University Research Journal-SURJ (Science Series) 2017, 49, 159-162.

Pauly, D. Some simple methods for the assessment of tropical fish stocks; Food & Agriculture Org.: 1983.

Sarker, B., Das, B., Chakraborty, S., Hossain, M. A., Alam, M. M., Mian, S., & Iqbal, M. M. (2022). Optimization of 17α-methyltestosterone dose to produce quality mono-sex Nile tilapia Oreochromis niloticus. Heliyon, 8(12), e12252.

Some specific comments

Point 2: Was the aim to develop or to optimise? Since there are already available methodologies for masculinization, I suggest to use the term optimise.

Response: Thank you for the excellent observation and suggestion. We have already revised the aim in the manuscript (Lines 38, 40 and 98).

Point 3: Any other welfare markers apart from body weight? 

Response: Thank you for the comment. Apart from the body weight, we have determined the male percentage and cost of treatments (Table 4). Furthermore, as per your suggestion, we have included a detailed analysis of the length-weight relationship and the relative condition factor in the revised manuscript (Lines 210-225; 351-365: and 446-451)

Point 4: Line 217: Either “below” or “lower” should be deleted

Response: Thank you for the suggestion. We have already made the changes on Line 242 in the revised manuscript.

Point 5: Lines 373-380. This part is confusing. How could the authors compare their results that are based on nanomaterials with conventional MT administration? Also, the 89,7% is low? Why do the authors mention “only”? I believe it should be rewritten

Response: Thank you for this excellent observation and already rewritten in the revised manuscript. We are optimizing the dose of MT with nanodelivery system. Therefore, we have compared the results of nanomaterials with conventional MT administration.

Lines 420-425: Importantly, red tilapia fry fed with T6 group produced 98.5% males, which is significantly higher in male tilapia percentage compared with the previous studies that found less than 70% sex-reversed males after feeding with 30 mg/kg of conventional MT administration for 28 days [6,57]. Our finding also confirmed that lipid nanoparticle could be a promising delivery system for androgen hormone.

Also, the 89,7% is low?

Yes, the 89.7% of males is low because farmers prefer all males (>98%) for grow-out farming. For tilapia grow-out farming to be profitable, this 10% of females could affect feeds, feeding, and fertilization. 

Why do the authors mention “only”? I believe it should be rewritten

We do apologize for the mistakes. The changes have been made in the revised manuscript.

Round 2

Reviewer 3 Report

The manuscript is improved and may be published in its current form